# Mid-infrared supermirrors with finesse exceeding 400 000

Gar-Wing Truong[1,6] ✉, Lukas W. Perner [2,3,6], D. Michelle Bailey [4], Georg Winkler[2], Seth B. Cataño-Lopez[1], Valentin J. Wittwer [5], Thomas Südmeyer[5], Catherine Nguyen[1], David Follman[1], Adam J. Fleisher [4], Oliver H. Heckl [2] ✉ & Garrett D. Cole [1]

For trace gas sensing and precision spectroscopy, optical cavities incorporating low-loss mirrors are indispensable for path length and optical intensity enhancement. Optical interference coatings in the visible and near-infrared (NIR) spectral regions have achieved total optical losses below 2 parts per million (ppm), enabling a cavity finesse in excess of 1 million. However, such advancements have been lacking in the mid-infrared (MIR), despite substantial scientific interest. Here, we demonstrate a significant breakthrough in high-performance MIR mirrors, reporting substrate-transferred single-crystal interference coatings capable of cavity finesse values from 200 000 to 400 000 near 4.5 μm, with excess optical losses (scatter and absorption) below 5 ppm. In a first proof-of-concept demonstration, we achieve the lowest noise-equivalent absorption in a linear cavity ring-down spectrometer normalized by cavity length. This substantial improvement in performance will unlock a rich variety of MIR applications for atmospheric transport and environmental sciences, detection of fugitive emissions, process gas monitoring, breath-gas analysis, and verification of biogenic fuels and plastics.

High-performance resonators integrating low-optical-loss mirrors are used in a wide variety of applications, ranging from time-and-frequency metrology, quantum electrodynamics and optomechanics, to trace gas sensing and detection. Rempe et al.[1] reported an ion-beam sputtered (IBS) coating at 850 nm on conventionally-polished substrates with transmittance $T = 0.5$ ppm and excess loss (consisting of scatter, $S$, and absorption, $A$) of $S + A = 1.1$ ppm. More recently, IBS coatings on micro-fabricated mirrors have shown even lower excess losses of $S + A = 0.74$ ppm (with $T = 1.9$ ppm) at 1550 nm[2,3]. In the MIR, only modest coating advancements have been realized and this technology significantly lags what is readily achievable in the NIR. The lowest reported excess loss for traditional amorphous MIR mirrors is $S + A = 30$ ppm with $T = 120$ ppm at 4.53 μm, resulting in a cavity finesse

of $\mathscr{F} = \pi\sqrt{R}/(1 - R) = 20\,900$[4], where $R = 1 - T - S - A$ is the reflectivity. Similar mirrors with reduced transmission values have yielded increased finesse values from 52 000 to 114 000[5–8], though the component losses ($T$ vs. $S + A$) in these systems are unknown. In contrast, substrate-transferred monocrystalline GaAs/AlGaAs Bragg mirrors at 4.54 μm, have shown the lowest levels of excess loss to date ($S + A = 7$ ppm)[9]. In that work, the transmission was high compared to the mirror losses, with $T = 144$ ppm, leading to a cavity finesse of 20 800, but with much greater on-resonance cavity transmission of $(T/(T + S + A))^2 = 91\%$ compared with 64% for previous results[4].

Here, we report on the realization of ultrahigh-finesse MIR mirrors using two coatings methods. The first method extends the recently-reported "all-crystalline" architecture[9] (Fig. 1a) to produce a

[1]Thorlabs Crystalline Solutions, 114 E Haley St., Suite G, Santa Barbara, CA 93101, USA. [2]Christian Doppler Laboratory for Mid-IR Spectroscopy and Semiconductor Optics, Faculty Center for Nano Structure Research, Faculty of Physics, University of Vienna, Boltzmanngasse 5, A-1090 Vienna, Austria. [3]Vienna Doctoral School in Physics, University of Vienna, Boltzmanngasse 5, A-1090 Vienna, Austria. [4]National Institute of Standards and Technology, 100 Bureau Drive, Gaithersburg, MD 20899, USA. [5]Laboratoire Temps-Fréquence, Institut de Physique, Université de Neuchâtel, Avenue de Bellevaux 51, 2000 Neuchâtel, Switzerland. [6]These authors contributed equally: Gar-Wing Truong, Lukas W. Perner. ✉e-mail: garwing@thorlabs.com; oliver.heckl@univie.ac.at

transmission-loss-dominated cavity finesse of 231 000 at 4.5 μm (on-resonance transmission >48%). A second "hybrid" approach yields even higher finesse, exceeding 400 000 (on-resonance transmission of 11%). This new hybrid coating paradigm (Fig. 1b) combines two different coating processes – an amorphous sub-stack deposition followed by bonding a crystalline multilayer on top. Owing to the minimal field penetration depth in quarter-wave reflective interference coatings (Fig. 2), these mirrors exhibit excess losses at least a factor of six lower than the previous best results obtained with all-amorphous interference coatings. This represents simultaneously the highest finesse and lowest excess losses yet achieved in this spectral region. The finesse and cavity transmission improvements enabled by these mirrors lead to compounding advantages for spectroscopy, resulting in record-low levels of noise-equivalent absorption. As a proof-of-concept demonstration, we present trace gas detection in ultra-high purity nitrogen.

## Results

### Design, fabrication, and optical validation

Figure 1 schematically shows the two low-loss mirror designs investigated in this manuscript. The all-crystalline structure consists of alternating high refractive index GaAs and low refractive index $Al_{0.92}Ga_{0.08}As$ layers. We additionally pursue the development of a hybrid mirror design that combines the same GaAs/AlGaAs multilayer with an IBS-deposited amorphous sub-stack. All mirrors use super-polished single-crystal Si substrates with a concave radius of curvature of 1 m, with the backside wedged at 0.5° and IBS-coated with a broadband 4–5 μm anti-reflection coating.

As described above, both mirror designs use a common crystalline "half-stack" multilayer grown via molecular beam epitaxy (MBE), comprising 22.25 periods of GaAs/$Al_{0.92}Ga_{0.08}As$ (22 × ¼-wave layer pairs, in terms of optical thickness, terminating with a ⅛-wave layer of high index GaAs). The epitaxial half-stack reduces the total physical thickness of the as-grown multilayer, minimizing the defect density and ultimately the optical losses of the crystalline coating. Deposition is carried out on a lattice-matched 15-cm-diameter [001]-oriented semi-insulating GaAs wafer. Following the crystal growth process, individual crystalline mirror die are defined via lithography and etching.

For the all-crystalline mirrors, two crystalline half-stack die are directly bonded to realize the full mirror stack, now 44.5 periods of GaAs/$Al_{0.92}Ga_{0.08}As$ with a nominal $T$ of 10 ppm at a target center wavelength of 4.5 μm. Next, the stacked epitaxial multilayer is transferred to the Si optical substrate, again using direct bonding with no adhesives or interlayers[10,11].

In the case of the hybrid mirrors, the base Si substrates are initially coated with an amorphous partial mirror structure via IBS. Amorphous Si (a-Si) is used for the high refractive index layers, with silica ($SiO_2$) as the low index component, with four periods of a-Si/$SiO_2$ deposited before a ⅛-wave terminating alumina ($Al_2O_3$) layer. This termination layer was intended to aid bonding, but subsequent designs ending the amorphous stack with a-Si were found to behave identically. An annealing process at 300 °C for 24 h minimizes internal stresses and reduces the optical absorption of the IBS films. Following the IBS deposition and annealing processes, we directly bond a GaAs/AlGaAs half mirror die to the capping $Al_2O_3$ layer to complete the hybrid mirror stack.

We explore both mirror designs to present the trade-offs between the all-crystalline mirrors (offering the lowest possible absorption loss) and the advantages of the hybrid approach (scalable manufacturing and access to longer center wavelengths). Given the optical field

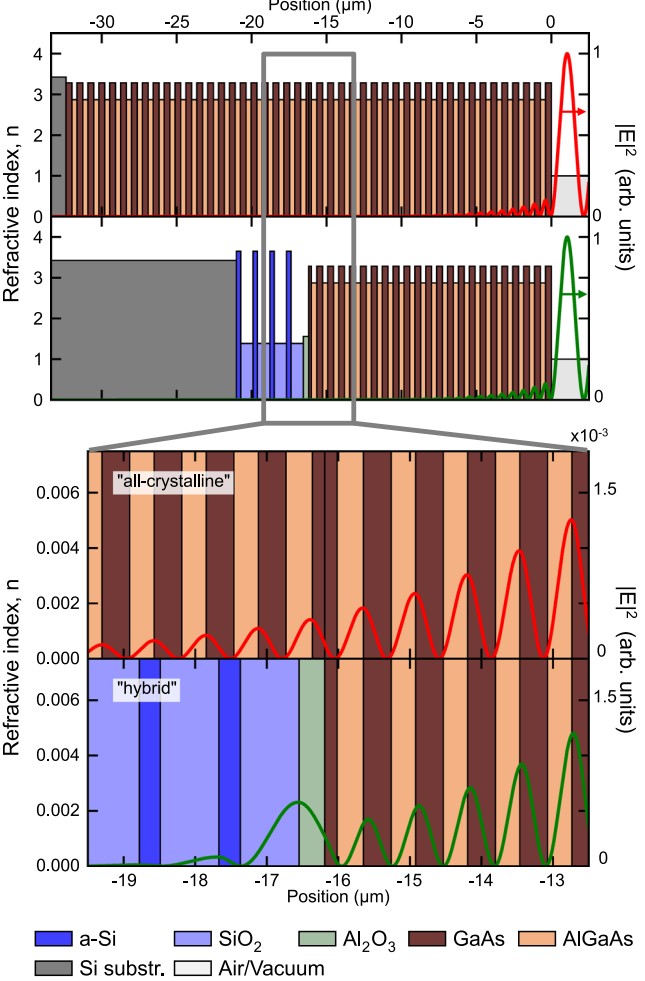

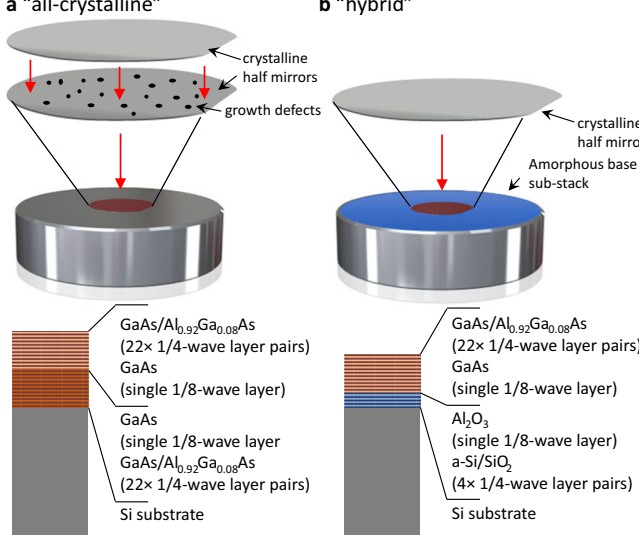

**Fig. 1 | Schematic description of the manufacturing process and coating structures.** These ultralow-loss mirrors leverage the low scatter and absorption losses of substrate-transferred MBE-grown GaAs/AlGaAs multilayers in (**a**) all-crystalline and (**b**) hybrid designs.

**Fig. 2 | Simulated decay of the electric field (red and green traces) as a function of depth into the all-crystalline and hybrid coatings.** The multilayer designs and material indices at 4.45 μm are provided. Replacement of one crystalline half-mirror with an amorphous stack has little impact on optical absorption if the crystalline layers remain the first reflection surfaces. Comparing the electric field profiles, the optical field penetration depth of 878 nm is identical for both the all-crystalline and hybrid coatings (for details see main text).

penetration depth of 878 nm for both the all-crystalline and hybrid coatings[12] compared to a total coating thickness of 32 μm and 21 μm for the respective designs, it is clear that the optical mode primarily samples the losses from the surface crystalline layers. A small amount of additional absorption of less than 5 ppm, conservatively estimated from worst-case material parameters, is expected from the more lossy amorphous stack due to the slightly enhanced field in the first couple of amorphous layers ($Al_2O_3$ and $SiO_2$) near the crystal-amorphous interface (Fig. 2). The hybrid coating approach also offers a wider stopband (due to the higher refractive index contrast of the amorphous layers), and the ability to extend the achievable center wavelengths, nominally to 12 μm and beyond.

In the hybrid mirror design, the epitaxial stack can be limited to a maximum physical thickness that maintains high structural perfection with limited growth-induced defects. A thinner crystalline multilayer (exhibiting minimal defects) enables ppm-level losses at wavelengths longer than that which can be achieved with the all-crystalline structure by avoiding excessively thick epilayers or successive bonding steps to push to wavelengths beyond 5 μm. Additional details of the manufacturing process and limitations are discussed in the Methods section.

High precision optical characterization of the completed mirrors to decompose the total loss into transmission and excess losses is possible with the all-crystalline coating since sample preparation required only cleaving of the epi structure. We followed the prescription given in ref. 13: i) X-ray diffraction (XRD) of the as-grown crystalline multilayer to determine the Al composition of the ternary $Al_xGa_{1-x}As$ layers; ii) cross-sectional scanning-electron microscopy (SEM) to probe the individual layer thicknesses for all materials; iii) Fourier-transform spectrometry (FTS) for the broadband transmittance spectra of the completed mirrors; iv) transmission matrix modeling (TMM) using the known refractive indices[13], layer

thicknesses, and broadband transmittance spectra to determine the transmittance within the stopband, where FTS provides insufficient dynamic range; v) finally, cavity ring-down gives the total loss, from which the transmittance can be subtracted to determine $S + A$. The uncertainty of $T$ at the stopband center ($T_O$) is estimated with a Monte-Carlo-style model, varying the measured TMM input parameters within their respective error bounds. Using the approach outlined above, calculations are performed for the all-crystalline mirror stack and the individual mirror transmission is found to be $T_O = 9.33 \pm 0.17$ ppm at $4449.5 \pm 0.5$ nm. A similar procedure was followed to derive the transmission of the hybrid coating by separately sectioning the amorphous sub-stack. We find $T_O = 2.53$ ppm at 4450 nm, with the evaluation of uncertainties left to a future study due to the increased difficulty in sample preparation of the insulating materials, as well as the appreciably higher complexity of the modeling for a 6-material system, rather than a 3-material system for the all-crystalline mirrors. Further information is provided in the Methods section.

The total loss measurements were performed using a cavity ring-down reflectometer implemented in a scheme based on that described in ref. 9 with added spectral and spatial scanning capabilities[14]. A broadband Fabry-Perot quantum cascade laser (FP-QCL) was mode-matched to the cavity under test with the mirrors separated by $145 \pm 2$ mm (calculated spot size of 629 μm on each mirror). Optical feedback from a grating in the Littrow configuration tunes the QCL, while a 35-dB optical isolator reduced unwanted optical feedback from the cavity that would dominate the laser behavior. The power transmitted through the cavity was filtered via a monochromator with a passband of 18 nm to provide spectrally resolved ring-down curves (admitting a weighted combination of up to ~260 longitudinal $TEM_{00}$ modes). During testing, the mirrors are held under vacuum (<1 mPa) to avoid intracavity atmospheric absorption.

Figure 3 shows the measured total loss near the region of minimum transmission for pairs of the all-crystalline and hybrid mirrors. For each data point, the center wavelength of the monochromator was set to the desired value and the Littrow grating angle was optimized for maximum optical transmission through the cavity and monochromator. Fewer valid ring-downs were obtainable towards the extremes of the laser tuning range, and the uncertainties increased due to a lower signal-to-noise ratio. Due to the free-running nature of the laser, we do not have an estimate of the statistical uncertainty of the laser wavelength.

The lowest total losses of $13.60 \pm 0.49$ ppm were observed with the monochromator set to 4.45 μm for the all-crystalline mirrors, corresponding to a finesse of $231\,000 \pm 8\,400$. These prototype mirrors typically showed a region of ~2 mm diameter over which these extreme levels of performance are maintained (see Fig. 4). Subtracting $T_O$, as determined by the TMM fit and XRD characterization of the stack, we infer an excess loss of $4.27 \pm 0.52$ ppm. If we assume scatter to be negligible and assign all excess losses in the GaAs/AlGaAs

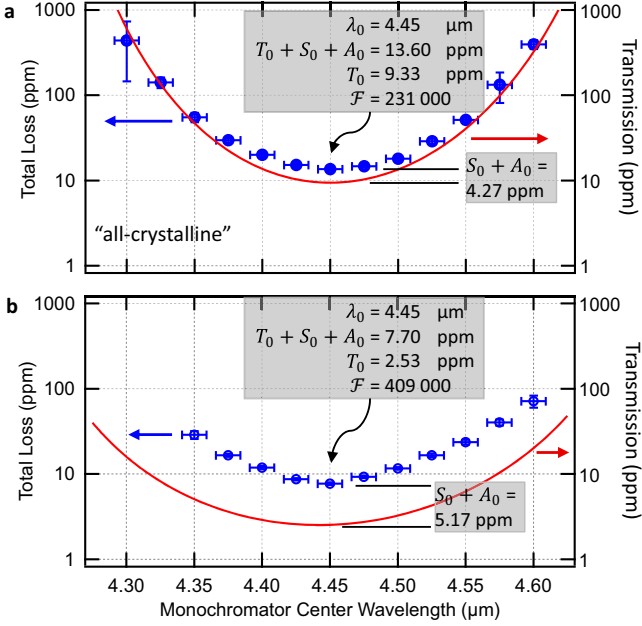

**Fig. 3 | Spectrally resolved measurements of optical losses.** Vertical error bars represent the standard deviation of the ensemble of ring-downs taken at a particular wavelength, whereas the horizontal error bars indicate the monochromator bandwidth of ±9 nm. Wavelengths at extremes of the laser tuning range resulted in lower source power and larger uncertainty in loss. Both mirror designs yield record optical performance, with (**a**) the all-crystalline mirrors achieving a transmission-dominated finesse > 200 000, while (**b**) the hybrid mirrors exceed 400 000 owing to a reduced transmission. Blue circles: measured total loss. Solid red: as-grown transmission ($T_{calc}$).

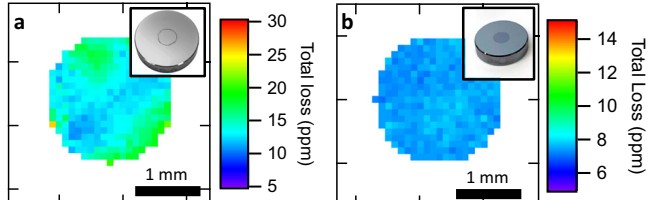

**Fig. 4 | Spatially resolved measurements of optical losses.** The value at each point was taken to be the median value of 10 consecutive measurements taken in the central 2 mm of (**a**) an all-crystalline mirror and (**b**) a hybrid mirror. Sample pitch was 0.1 mm. This measurement reveals a significant enhancement in uniformity (smaller range of loss values) for the hybrid mirrors, likely given the reduced defect density from the single bonding step. Photos of example mirrors are shown in the insets.

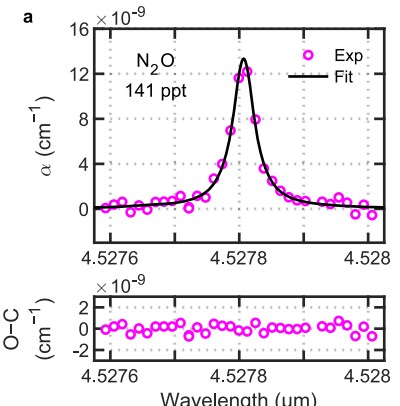

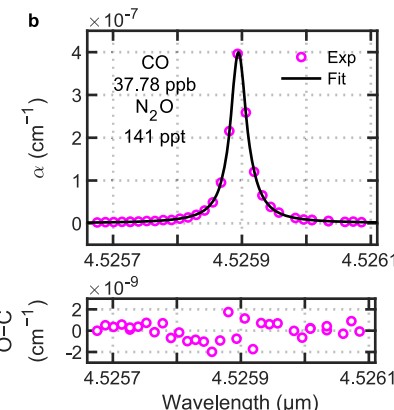

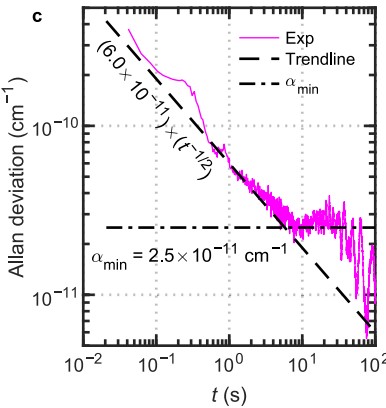

**Fig. 5 | Cavity ring-down spectroscopy of an ultra-high purity nitrogen (N₂) sample.** Absorption lines from trace impurities of (**a**) nitrous oxide (N₂O) were consistent with a concentration of 141 ppt and (**b**) carbon monoxide (CO) at 37.78 ppb (with a minor contribution from N₂O). Abbreviation: O−C, observed minus calculated. (**c**) Allan deviation of empty-cavity time constants ($\tau_0$).

multilayer to absorption, this corresponds to an absorption coefficient, $\alpha_{coat}$, of 0.0246 cm⁻¹ ± 0.0030 cm⁻¹, and an extinction coefficient of $k_{coat} = (8.7 \pm 1.1) \times 10^{-7}$ for the coating.

The lowest losses observed for the hybrid mirrors was 7.70 ± 0.27 ppm, corresponding to an even higher finesse of 409 000 ± 14 000, which represents nearly an order of magnitude increase over any reported conventional coating in the MIR. Again, subtracting $T_0$ (determined for the hybrid coating structure), we observe a similar level of $S + A = 5.17$ ppm. This is consistent with the all-crystalline results given that the field penetration depth is substantially shallower than the thickness of the epitaxial multilayer.

It is known from ref. 9 that MIR crystalline coatings can exhibit polarization-dependent absorption. We performed an empirical verification by rotating both the laser polarization as well as the coating axes. Within the precision of our ring-down reflectometer, we observed no change in loss with laser and mirror rotation for both types of mirrors. It is not yet clear why these stacked coatings show little to no orientation-dependent losses and further investigations are ongoing.

By successively repositioning the mirrors (and realigning to ensure the curved surface of the mirrors remain normal to the laser), it was possible to map the optical losses across the coating[14]. Figure 4 shows the measured losses in the central 2 mm for a representative unit of each mirror type. Variations are obvious on the all-crystalline mirror, and we attribute these to the combined imperfections from the double-bonding process. It is evident that the hybrid mirror greatly improves uniformity by avoiding the additional epitaxial growth defects from a second half-stack, as well as from the reduced complexity of the single bonding step.

Given their improved uniformity and low losses, we explored the performance of the hybrid mirrors in a cavity ring-down apparatus at the National Institute of Standard and Technology (NIST), Gaithersburg, optimized for trace gas sensing and quantification.

## Applications in gas sensing and spectroscopy

A MIR cavity of unprecedented finesse offers intriguing possibilities for improving the sensitivity of instruments that are widely used for quantitative trace gas sensing. Here, we present a proof-of-concept demonstration of cavity ring-down spectroscopy using a pair of hybrid mirrors forming a 79-cm long cavity. To search for trace amounts of molecular absorption, we flowed a sample of ultra-high purity nitrogen (N₂; 99.999%) through the cavity at a pressure of 13 kPa and a temperature of 297 K. Similar ultra-high purity gases are used in precision gas mixing as balance gases and as feed gases in the semiconductor industry, where purity is of paramount importance in achieving high

yield. Figure 5a shows an isolated absorption line from residual nitrous oxide (N₂O), detected at an amount fraction of 141 ppt ± 3 ppt. In a nearby spectral window, Fig. 5b shows an absorption line from residual carbon monoxide (CO), detected at an amount fraction of 37.78 ppb ± 0.12 ppb (N₂O also makes a minor contribution). These observed amounts are two to three orders-of-magnitude below atmospheric abundances for N₂O and CO.

Separately, we measured the Allan deviation of the empty-cavity absorption coefficient ($\alpha_0$) at a fixed wavelength of 4.527 μm (Fig. 5c) and determined a minimum absorption coefficient of $\alpha_{min} = 2.5 \times 10^{-11}$ cm⁻¹ achieved at 8 s. For timescales below 8 s, the Allan deviation approximately follows a $t^{-1/2}$ dependency (dashed trendline), indicating that measurements were dominated by white noise, before being affected by drifts. The value of $\alpha_{min}$ corresponds to a noise-equivalent concentration for N₂O and CO of 0.3 ppt and 2.2 ppt, respectively. These projected limits—established here by proof-of-principle spectroscopy using mirrors fabricated with our breakthrough hybrid MIR coating technology—are substantially better than the sensitivity metrics of existing commercial optical analyzers that operate at similar pressures and temperatures in the same wavelength region (e.g., ≥25 ppt N₂O-in-air at 8 s for a sample of 326.5 ppb N₂O-in-air; estimated using[15]). The 4.5 μm spectral region is also of great interest to optical detection of radiocarbon dioxide (¹⁴CO₂)[5,16–19], where increased cavity performance can also be of benefit.

In Table 1 we compare our Allan deviation results at 1 s from Fig. 5c to other values found in the literature[17,20–24]. We compare across similar realizations of linear-absorption cavity ring-down spectroscopy at wavelengths near 4.5 μm where the laser and the cavity are passively coupled, e.g., excluding results that leverage optical feedback self-locking[25] to substantially increase throughput[22]. Also not included are

## Table 1 | Performance of comparable cavity ring-down spectrometers operating near 4.5 μm

| Reference | $\alpha_0$ at 1 s (cm⁻¹ Hz⁻¹/²) | $L$ (m) | $\alpha_0$ at 1 s for $L$ = 10 cm (cm⁻¹ Hz⁻¹/²) |
|---|---|---|---|
| INO-CNR[20] | 5.0 × 10⁻⁹ | 1 | 5.0 × 10⁻⁷ |
| VTT[21] | 2.1 × 10⁻⁹ | 0.4 | 3.4 × 10⁻⁸ |
| Nagoya[22] | 1.1 × 10⁻⁸ | 0.11 | 1.3 × 10⁻⁸ |
| NIST[23] | 2.6 × 10⁻¹¹ | 1.5 | 5.9 × 10⁻⁹ |
| LLNL[17,24] | 1.2 × 10⁻¹⁰ | 0.67 | 5.3 × 10⁻⁹ |
| This work | 6.0 × 10⁻¹¹ | 0.79 | 3.7 × 10⁻⁹ |

Here the absorption coefficient at 1 s integration, cavity length, and normalized performance for each system are shown. The ultralow optical loss of our crystalline coatings yields the lowest normalized noise-equivalent absorption.

comparisons to non-linear cavity ring-down spectroscopy techniques[7,16,18,26,27]. To level the comparison with respect to at least one instrumental variable, we normalize the literature values to a common cavity length of 10 cm (assuming that photodetection is detector-noise limited[28] so that the noise equivalent absorption, α at 1 s, is inversely proportional to the square of the cavity length according to $\alpha_0 = (1/(c\tau_0^2))\sigma_{\tau_0}$, where $c$ is the speed of light, $\tau_0 = \mathcal{F}L/(c\pi)$ is the empty cavity decay time, $\mathcal{F}$ is the empty cavity finesse, and $\sigma_{\tau_0}$ is the standard error on $\tau_0$ at 1 s. While some of the groups whose results appear in Table 1 have published more recent papers that report modified or applied instrumentation, to the best of our knowledge, we list here the lowest MIR (4.5 μm) 1-s absorption coefficients published to date.

## Discussion

The MIR region has been recognized as a rich portion of the optical spectrum where many species of interest absorb strongly and characteristically. Relative to the rapid maturation of laser source technology (such as QCLs), low loss interference coatings based on conventional deposition techniques have now become the performance-limiting elements of instruments such as cavity ring-down spectrometers. Here we demonstrate a new paradigm for MIR coatings using crystalline GaAs/AlGaAs multilayers in two related, but distinct methodologies. These fundamentally new ultra-high reflectivity coatings are shown to achieve record levels of finesse values in a linear cavity configuration. These ultra-high finesse values have been achieved with record-low excess losses, in combination with comparable transmission values (and being transmission-loss dominated in the case of the all-crystalline mirrors). These metrics are key for applications in which cavity transmission and contrast are paramount.

While the all-crystalline mirrors yield phenomenal results, the double-bonding of the half stacks is a challenging process that introduces roadblocks to scalability in terms of manufacturing and access to longer center wavelengths. As a promising alternative, we demonstrate a hybrid mirror structure combining amorphous multilayers deposited with traditional physical vapor deposition (PVD) processes, capped with a substrate-transferred single-crystal stack. This platform allows for easier fabrication of low-loss MIR mirrors and further extends the operating wavelength of these coatings to the long wavelength limit set by optical phonon absorption in the reststrahlen band beyond approximately 20 μm in GaAs[29]. This platform should allow us to realize ppm-levels of excess losses (<10 ppm) out to ~7000 nm and <50 ppm for wavelengths approaching 11 μm (enabling, for the first time, high-performance $CO_2$ laser optics). Long-wavelength interference coatings capable of ppm-levels of optical losses have the potential to advance capabilities in laser-based manufacturing, including extreme-UV lithography.

We anticipate that such coatings will enable spectroscopy at sensitivity levels previously unattainable with existing technologies, furthering tabletop radiocarbon spectrometry[4–6], compact and fieldable atmospheric trace gas detection devices, and chemical sensing instruments[30–32]. Furthermore, the presented mirrors have the potential to significantly advance the capability of high damage threshold[33,34] and low-loss optics for long-wavelength laser-based manufacturing systems.

## Methods
### Sample fabrication
A single epi wafer was used for the interference coating, with direct bonding of two half-stack multilayer structures carried out at the die level to build-up the Bragg reflector in the all-crystalline mirrors. This initial bonding step is employed to minimize the density of growth defects in these relatively thick MIR coatings. The advantage is two-fold: i) the shorter growth process for the half mirror minimizes defect incorporation and lowers the overall defect density in the thin films,

and ii) defects terminating at the surface of the as-grown crystal are mitigated as they are buried at the bond interface in the middle of the combined stack. Following the initial bonding process, a substrate and etch stop removal process is carried out via selective wet chemical etching, exposing the back surface of one of the half mirror stacks. This exposed face of the epi structure is then directly bonded to a super-polished silicon optical substrate (1 m concave radius of curvature, 6.35 mm thick with a 25.4 mm outer diameter, backside wedged at 0.5°) using a room temperature plasma-activated bonding process. To complete the all-crystalline MIR supermirrors, a second substrate and etch stop removal process is carried out, leaving the stacked crystalline coating transferred to the Si base substrate. Given the need for very thick stacks in the all-crystalline approach, mirrors with a transmission below 10 ppm would require additional stacking/bonding steps for wavelengths beyond ~5.5 μm (assuming a maximum epitaxial growth thickness of ~16 μm for maintaining a suitably low defect density below 1000 cm$^{-2}$). Typical RMS surface roughness below σ = 0.2 nm is achieved[9], which leads to negligible expected scatter levels of order $1 - Exp(-(4\pi\sigma/\lambda)^2) \approx 0.3$ ppm[35].

With the hybrid approach, the epitaxial stack can be limited to this maximum thickness value and the underlying amorphous structure can be tailored to tune the transmission. The die level "stacking" process is skipped and the GaAs/AlGaAs disk is instead directly bonded to the surface of a pre-coated Si base wafer. The pre-coating is a partial high-reflectivity stack consisting of the amorphous a-Si/SiO$_2$ layers terminated with a $\frac{1}{8}$-wave Al$_2$O$_3$ layer. These amorphous films are generated via IBS in a Navigator 1100 (CEC GmbH) coating system using xenon as a sputtering gas. The oxides (SiO$_2$ and Al$_2$O$_3$) are formed by oxidation of metallic Al (>99.999% purity) and Si (p-doped >99.9999% purity) released from the sputtering targets and adding 70 sccm oxygen to the process chamber for SiO$_2$ and 90 sccm for Al$_2$O$_3$. The a-Si was deposited from a Si (undoped, 99.9999% purity) target without adding any oxygen to the chamber. Before deposition, the vacuum chamber is evacuated to approximately $1 \times 10^{-5}$ Pa. During deposition, the vacuum pressure did not exceed 0.2 Pa and the substrate holder was heated and temperature controlled to 150 °C. In a first run the a-Si/SiO$_2$ layers were deposited followed by Al$_2$O$_3$ deposition without breaking the vacuum or turning off the plasma source. The entire coating run was controlled and monitored with a broadband optical monitoring system (wavelength range of 250 nm to 2200 nm) on an IR fused silica monitoring glass. After deposition, the films are annealed for 24 hours at 300 °C in an oven at ambient atmosphere (stainless steel hot air oven, Memmert GmbH).

These prototype MIR mirrors employ 8 mm diameter epitaxial coating discs. This small crystalline coating area was used to conserve material during test mirror production. It is important to note that we can routinely produce much larger crystalline mirrors, with high-performance 10-cm diameter crystalline coatings recently demonstrated[36].

### Coating transmission determination
To accurately determine the minimum transmission value for each mirror, we must determine the refractive indices and physical thicknesses of each layer. As the all-crystalline mirrors consist of only two materials, the process is simpler. In this case, we first determine the exact alloy composition of the low index Al$_x$Ga$_{1-x}$As layers, then extract the thicknesses of the films in the as-grown multilayer structure. XRD is employed to determine the AlAs mole fraction, $x$, in the ternary Al$_x$Ga$_{1-x}$As alloy as $x = 92.9\%$ in the as-grown 22.25-period epitaxial half-stack material. Individual layer thicknesses were determined via SEM imaging of a cleaved sample of the MBE-grown multilayer. These were evaluated line-by-line, resulting in a large sample size that drastically improved the statistical uncertainty of the thickness determinations (the exact process is described in ref. 13). However, it is known that deposition rate variations across the MBE platen will lead to modest

differences in the absolute layer thicknesses across the wafer (or across various wafers dependent on the growth configuration), while the ratio of thicknesses are maintained[37]. That is, we expect the layer thicknesses in our mirrors to be related to the cleaved sample by a single rescaling parameter, i.e., by multiplying the obtained thickness values by a single factor $d_{scale}$.

These unavoidable variations in alloy fractions and thickness in the MBE growth process led to a deviation of the actual transmittance spectrum from the design target. To obtain an accurate transmittance $T$ spectrum over the stopband region, including the region of lowest losses around the mirror center wavelength $\lambda_0$, we employed an improved variant of the approach presented in ref. 9. We recorded a transmission spectrum of the all-crystalline mirrors with a Fourier-transform Spectrometer (FTS, Bruker VERTEX 80 v). An example spectrum is shown in Supplemental Fig. 1. These broadband measurements were recorded in vacuum (220 Pa) with a resolution of 0.5 cm⁻¹. Due to the limited sensitivity and low signal-to-noise ratio (SNR) in the spectral region around $\lambda_0$, the transmittance cannot be directly probed. However, the width of the stopband and the characteristic structure of the sidelobes allow for a precise interpolation if the thickness of the layers are accurately known. Knowing both, the broadband transmittance spectrum and the physical layer thicknesses allows us to fit a transfer-matrix method (TMM) model to the FTS data to obtain accurate transmittance values in the stopband region. In this curve-fitting exercise, we modeled the layer structure using refractive index values measured in-house for the GaAs and AlGaAs layers using the cleaved sample of epitaxial material grown in the same run as for the completed mirrors (see ref. 13 for details), while for the Si substrate we used values from ref. 38, and $n = 1$ for incidence and exit media. Note that we assumed abrupt interfaces and no variation in AlAs mole fraction throughout the structure, as well as a perfect AR coating on the back surface of the substrate.

The subsequent fit routine had three free parameters: The aforementioned multiplicative scaling factor $d_{scale}$ on layer thicknesses, a global scaling parameter $T_{scale}$ to counteract photometric inaccuracies in FTS imaging due to geometric effects (mostly caused by the relatively thick, wedged, and highly refractive Si substrate), and finally a single scaling parameter for the middle GaAs layer (formed of the two $\frac{1}{8}$-wave caps of the half-mirrors) to counteract a systematic error in SEM imaging. The measured film parameters were varied within their respective error bounds using a Monte-Carlo-style approach to obtain an estimate of the uncertainty of the transmittance at the stopband center $T_0$.

For the hybrid mirrors, the crystalline stack came from the same epi wafer as employed for the all-crystalline mirrors, thus those findings (in terms of physical thicknesses and refractive indices) can be directly used. The challenge in the case of these structures, now consisting of six materials instead of two, is the fact that the amorphous materials have less well-defined refractive index values. Thus, we undertake a similar systematic study of the amorphous materials, including ellipsometry (SENresearch 4.0 and SENDIRA, SENTECH Instruments GmbH) on single films deposited and processed under identical conditions, photometric measurements in the NIR and MIR (Agilent Cary5000, ThermoFisher Nicolet iS50) on witness pieces and single films, etc. in order to estimate the transmission of the completed hybrid mirror. In this case, larger error bounds remain for the refractive indices and physical thicknesses of the amorphous films and we will pursue more in-depth analysis of these materials in follow-on work. Additionally, advances in epitaxy exploiting alternative material platforms could enable wider bandwidth mirrors via enhanced index contrast[39]. However, it remains to be seen whether such material systems will be capable of sufficiently low optical loss. The higher index contrast achievable with the amorphous sub-stack resulted in a wider stopband than the all-crystalline coating, as determined by the wavelength range over which the calculated transmission increased by a factor of two (approximately 180 nm and 130 nm for the hybrid and all-crystalline coatings, respectively).

## Ringdown spectroscopy measurements

To perform spectroscopy, a continuous-wave distributed feedback quantum cascade laser (DFB-QCL) is tuned in steps of 0.013 nm (equal to the cavity free-spectral range, $\nu_{fsr} = 189$ MHz) around 4.52 μm and cavity transmission is measured on a cryogen-cooled InSb detector, amplified, and digitized for analysis. An acousto-optic switch is used to shutter the laser pumping of the cavity when a predetermined transmission threshold is met and then the optical decay is recorded. The empty-cavity time constant was $\tau_0 = 128$ μs which, for a cavity length of 79 cm, equates to a per-mirror loss of $1 - R = 21$ ppm − a value consistent with the data plotted in Fig. 3. An example of a raw time-domain optical decays is shown in Supplementary Fig. 2.

The cavity construction rigidly mounted the hybrid mirrors within a flow cell, using Invar rods to mitigate against drifts in cavity length due to variations in laboratory temperature. The gas sample is introduced as a continuous flow at a nominal rate of 20 standard cubic centimeters per minute (sccm) and pressure of 13 kPa. The pressure is stabilized using a back pressure controller and monitored by pressure gauges calibrated to NIST secondary standards. The temperature of the outer walls of the flow cell−assumed equal to the temperature of the sample gas−is measured by two NIST-calibrated platinum resistance thermometers to be 297 K.

The amount fractions of molecular impurities found in the ultra-high purity $N_2$ gas sample are determined by fitting Voigt line shape functions to the measured spectra using parameters from the HITRAN2020 database[40] as initial guesses. The temperature and pressure of the gas sample are fixed to the measured experimental values and the following parameters are floated in each fit: an absolute frequency offset, $N_2O$ or CO amount fraction and a linear baseline model for $\tau_0$. For the spectrum containing CO absorption, the $N_2O$ amount fraction is fixed to the fitted value determined from the isolated line, and the following two additional parameters are floated: pressure-broadening coefficient for CO and relative frequency between the CO (strong) and $N_2O$ (weak) transitions.

The calculation of noise-equivalent amount fractions for $N_2O$ and CO is done for the transitions shown in Fig. 5a, b using the HITRAN2020 database[40] and a Voigt line shape function modeled at a pressure of 5 kPa and a temperature of 296 K − pressure and temperature conditions commonly used in commercial sensors like those reported in ref. 15.

## Reporting summary

Further information on research design is available in the Nature Portfolio Reporting Summary linked to this article.

## Data availability

All data generated in this study in relation to displayed figures have been deposited in the NIST database at https://doi.org/10.18434/mds2-3089.

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

## Acknowledgements

We thank C. Manning, A. Helbers, and M. Philippou for an inhouse FP-QCL monitoring solution; Y. Dikmelik for the FP-QCL source; J.T. Hodges and D. Mazzotti for manuscript review. Funding sources include Thorlabs internal (GWT, LWP, GW, SBCL, CN, DF, OHH, GDC), NIST internal (DMB, AJF), National Foundation for Research, Technology and Development, Austrian Science Fund, FWF, (P 36040) (LWP, GW, OHH), Schweizerischer Nationalfonds zur Förderung der Wissenschaftlichen Forschung (SNSF) (206021_198176) (VJW, TS). A portion of this work was performed in the UCSB Nanofabrication Facility, an open access laboratory, and at the Faculty Center for Nano Structure Research at the University of Vienna. The financial support by the Austrian Federal Ministry for Labour and Economy and the National Foundation for Research, Technology and Development and the Christian Doppler Research Association is gratefully acknowledged.

## Author contributions

G.W.T. and L.W.P. contributed equally to this work. G.W.T., L.W.P., G.W., and S.B.C.L. measured the mirror performance. G.W.T., L.W.P., and V.W. characterized and modeled the crystalline and amorphous mirror materials. G.D.C. designed the interference coatings and C.N., G.D.C.,

and D.F. performed the crystalline coating process. V.W. co-designed the hybrid mirrors and deposited the amorphous multilayer structures via IBS under the supervision of T.S. D.M.B. and A.J.F. performed the spectroscopic measurements. O.H.H. and G.D.C. provided technical oversight and experimental design. All authors contributed to the manuscript.

## Competing interests

Thorlabs, Inc holds patents (U.S. Patent 9,945,996/E.U. Patent 2607935A1, E.U. Patent 3219832A1, U.S. Patent 11,365,492/E.U. Patent 3219834A1, U.S. App. # 17/683,776) related to this work. The other authors declare no competing interests. Certain instruments are identified in this paper in order to specify the experimental procedure adequately. Such identification is not intended to imply recommendation or endorsement by NIST, nor is it intended to imply that the instruments identified are necessarily the best available for the purpose.
