## [Peer Review File · Nature Communications]

Mid-infrared supermirrors with finesse exceeding 400 000REVIEWER COMMENTS

Reviewer #1 (Remarks to the Author):

Review for "Transmission-dominated mid-infrared supermirrors with finesse exceeding 200 000" by Truong et al. (2022)

In their manuscript, Truong et al. report on extremely high quality mirrors at 4.45 μm central wavelength and a cavity built using these. Keeping the requirements for realizing useful optical cavities (which probably will be the main application) in mind, the transmission is chosen to be larger than the scattering/absorption losses of the coatings, and this choice is also reproduced in the device. The reported spectral region is, as the authors claim, interesting for spectroscopy but optics in this spectral region are not yet at the standard of optics in the visible and near-infrared.

The manuscript starts with a good overview of comparable work and the current state of the art to compare the achieved parameters. The reported mirror coatings are of the highest quality and the employed characterization methods are all adequate. The presented data supports the claims in the manuscript.

What I lack from reading the manuscript (and think other readers might be interested in this, too) is a clear statement what is the main reason for the reported leap in performance:

- If it is the stacking of two half-mirrors, would increasing the number of stackings and decreasing the number of grown layers further help?
- If it is growth, what do I need to adapt to achieve this level of quality?
- If it is the design, i.e., the chosen transmission versus the losses, are the chosen values the best possibility or how would I choose even better?

I believe such a discussion would be interesting and separate the novel aspects of the presented devices/fabrication process over the incremental improvements.

The report is technical and brief, therefore, it requires a level of knowledge of the field higher than I would assume for the general reader of a multidisciplinary journal such as Nature Communications. Nevertheless, when addressing the above point and adding some methodological explanations, I think it can be considered for publication in Nature Communications.

Minor comments:

- 1) Fig 1: The top horizontal axis lacks its label.
- 2) A visualization of the fabrication process could be helpful. It is currently described but seems to be one of the reasons for the achieved high quality of the mirrors. Therefore, it could deserve a figure.
- 3) Fig. 3 shows vertical error bars only for 2 data points. I assume the other error bars are smaller than the data point markers. If this assumption is true, the marker size could be decreased. Also, the authors could comment on why these two data points show much larger uncertainties.
- 4) The authors performed cross-sectional SEM to probe layer thicknesses but show no data. I would be interested if any differences are visible between the purely grown and the bonded layers in the middle of the stack.
- 5) I don't understand the reference on page 2: (see Figure 1); v)
- 6) The paper says the spatially dependent data in Fig. 4 was taken by translating the cavity mirrors with respect to the incoming beam and attributes the observed variations to the bonding processes. Could translation away from the optical axis also cause a misalignment of the cavity? I would appreciate a comment on this aspect.
- 7) The methods section could be subdivided.

Reviewer #2 (Remarks to the Author):

The work reported the world-record finesse of 200,000 on a pair of infrared mirrors at 4.45 μm , which is a significant advancement in the mid-infrared band and could improve the performance of cavity ring-down spectroscopy for trace gas sensing by orders of magnitude. It is also impressive to see that the quality and uniformity of the epitaxy-grown coating has been quickly approaching its counterparts in the visible and near-infrared bands. I believe that the significance of this result has reached the standard for publication in Nature Communications, although the technique for mirror fabrication has already been published before. The experimental methods are solid, and the manuscript is well written. Before accepting, I would like to make the following suggestions and comments.

1. In Fig. 2(a), the cavity ring-down traces seem to start from quite different signal levels. Is it because of the fluctuation of the input power to the cavity or something else fluctuating in the setup?
2. In the first paragraph of page 2, where the coating design has been discussed, can you provide some quantitative information on surface and bulk scatters? For example, surface roughness measurements on top surfaces of the end mirror or the half mirror stack that could provide an estimate on scattering loss.
3. Relating to the comment above, in the first paragraph of page 3, where the linear absorption coefficient has been calculated, how does the result compare with the absorption coefficient of epitaxy-grown GaAs and AlGaAs films reported in the literature? I think it would be beneficial to compare these results to confirm the absorption is the limiting factor here. And more importantly, is there a path towards even lower excess optical loss of these mirrors?
4. As a minor comment, in the third paragraph of Methods, I realize that the refractive indices of the GaAs and AlGaAs layers was measured on a cleaved sample in the same epitaxy run after several reads. Is that correct? But when first read, the sentence "we modeled the DBR structure using refractive index values measured in-house for the GaAs and AlGaAs" does not seem to specify whether the measurement was done on the bonded mirror or the cleave sample. It would be beneficial to point this out for the general audience.

Reviewer #3 (Remarks to the Author):

The manuscript "Transmission-dominated mid-infrared supermirrors with finesse exceeding 200 000" by Truong et al. reports the production and careful characterization of high-reflectivity and low-optical-loss mirrors for mid-infrared wavenelgths around 4.45 μm .

The paper is well-written and concisely conveys the main results. These results are indeed impressive and bare the potential to advance the applications mentioned in the manuscript, and should therefore be published. However, as it stands, the manuscript would be suitable for publication in a journal specialized in optics rather than in a journal aimed at a broad readership, such as Nature Communications.

The authors could, however, substantially widen the scope of the paper by adding a proof of principle, demonstrating, e.g., that indeed this cavity can improve detection limits for trace gases by orders of magnitude as compared to the state of the art. In this case, I believe that this work could fit to the scope of Nature Communications.

In addition, addressing the following points could strengthen the paper:

1. The claim "... allowing improvements in detection limits by orders of magnitude." from the abstract is not clearly substantiated in this paper. For instance, the improvement in cavity finesse with respect to [5] is "only" a factor of 4. Even though the authors comment that in refs. [5,6] the component losses are not stated, a more conservative claim in the abstract would be more accurate.
2. The bandwidth of the mirrors could be described more precisely in the text of the paper.
3. On page 2, I do not understand the sentence "Mirror die were defined via lithography and etching..."
4. Importantly, for a paper addressing a broad audience, the state of the art should be presented in a broader context. The authors cite only 18 references, 7 of which are to their previous work. It would improve the quality of the paper if they could, e.g., give an overview of resonators demonstrated in the mid-infrared spectral region.

Reviewer #4 (Remarks to the Author):

The paper by Truong and co-workers introduces for the first time mirrors with extremely high finesse at mid-infrared wavelengths, overcoming the well-known reflectivity and loss limitations in this wavelength range. The result obtained, namely a finesse of 230000 at 4.45 μm with only 4 ppm losses, corresponding to a 45% cavity throughput under optimal conditions, sets for these mirrors a quality on a par with near-infrared mirrors. In the long term, this is likely to produce important advances in precision spectroscopy and ultra-sensitive gas detection.

It is very difficult to decide whether this paper deserves or not a publication in Nature Communications: on the one hand the quality of the result and its scientific relevance, particularly if these mirrors could be made commercially available in a future for applications, would definitely justify a high-impact factor publication. On the other hand, this paper does not contain any novelty as compared to the Optica paper published two years ago by the same authors: the technology is the same, the methods are nearly the same, the mirror excess losses are just less than a factor of 2 better (4 ppm rather than 7 ppm). What changes is the number of periods in the mirror stack, increased from 34 to 44. This technical step, though it was not obvious, was already well within the expectations of anyone that had read the first paper (even the magic number of periods, 44, was already anticipated in the conclusions of the Optica paper). I let the Editor take a final decision on the basis of this and other reviews. If this paper is accepted for publication, I just encourage the authors to better clarify these points:

- The total losses measured with cavity-ring down spectroscopy are given with error bars that seem to be overestimated. For example, at the optimal wavelength of 4.45 μm , the total losses measured by averaging 500 ring-down signals are said to be 13.6 ppm \pm 0.49 ppm, which is equivalent to an uncertainty of 3.6%. This would imply, for a single ring-down event, an uncertainty as large as $3.6\% * (500)^{0.5} = 80\%$. The authors should then better comment the origin of their error bars.
- The cavity-ring-down traces in Fig.1 show a pretty poor signal-to noise ratio because the peak-to-peak residuals (10 mV, panel a) are only 10-15 times below the maximum signal (100-150 mV, again panel a). Why is the signal-to-noise ratio so poor ?
- The number of ring-down signals N that has been used to determine the mirror losses is strangely kept very high around 4.45 μm where the ring-down time and thus the SNR is maximum (N about 500), while being kept low at the edges of the mirror reflection band (N=60 at 4.33 μm , when the ring-down time is 10 times smaller). Shouldn't one choose the opposite approach, with a higher N at the edge of the reflection bandwidth?
- I would like to have a comment on the spectral width of the reflection band: this is actually

pretty narrow if one considers that the reflection drops by a factor of 10 at 4.33 and 4.58 μm , which are only 5% away from the maximum reflectivity point. How could one enlarge the reflection band ?

Reviewer #1 (Remarks to the Author):

Review for "Transmission-dominated mid-infrared supermirrors with finesse exceeding 200 000" by Truong et al. (2022)

In their manuscript, Truong et al. report on extremely high quality mirrors at 4.45 μm central wavelength and a cavity built using these. Keeping the requirements for realizing useful optical cavities (which probably will be the main application) in mind, the transmission is chosen to be larger than the scattering/absorption losses of the coatings, and this choice is also reproduced in the device. The reported spectral region is, as the authors claim, interesting for spectroscopy but optics in this spectral region are not yet at the standard of optics in the visible and near-infrared.

The manuscript starts with a good overview of comparable work and the current state of the art to compare the achieved parameters. The reported mirror coatings are of the highest quality and the employed characterization methods are all adequate. The presented data supports the claims in the manuscript.

We thank the reviewer for their very positive feedback on our submitted manuscript.

What I lack from reading the manuscript (and think other readers might be interested in this, too) is a clear statement what is the main reason for the reported leap in performance:

- If it is the stacking of two half-mirrors, would increasing the number of stackings and decreasing the number of grown layers further help?

The stacking process utilized here indeed improves the losses by significantly reducing the limiting scatter loss in the coatings [Cole 2016, Bjork 2017]. However, there is a balance between probability of stacking-related and growth-related defects. The all-crystalline production process is quite complex, requiring two bonding steps to generate a single completed mirror (half DBR stacking followed by substrate transfer). Although further improvements could potentially be realized by stacking more partial mirror structures, the required number of bonding processes would become impractical. Thus, in addition to the double-bonded all-crystalline mirrors, we additionally explore a hybrid approach that exploits a base amorphous structure capped with a crystalline multilayer.

In adding the hybrid coating concept to the revised manuscript, we describe the pros and cons of both approaches which covers the topics above (in the "Design, Fabrication, and Optical Validation" and "Methods" sections).

- If it is growth, what do I need to adapt to achieve this level of quality?

It is critical to minimize the background doping (impurity) level in the deposited layers. Secondly, it is important to realize abrupt interfaces and minimal surface roughness. Quantifying these properties, we have realized background doping at the $\leq 10^{14} \text{ cm}^{-2}$ (acceptor dominated), and surface micro-roughness values of $\sim 0.13 \text{ nm}$ (measured with over a $10 \mu\text{m} \times 10 \mu\text{m}$ area of the coating using an atomic force microscope). This work is a direct extension of our efforts on mirrors for reference cavities for

ultrastable laser systems [Cole 2013]. We have worked diligently over the last ~10 years to further improve the crystalline coating quality to realize performance on-par with the state-of-the-art ion-beam sputtered multilayers; some details being covered previously [Cole 2016]. With an optimized molecular beam epitaxy (MBE) process now defined, ppm-levels of optical loss are now readily achievable in the near- and mid-infrared spectral regions. For mirrors with center wavelengths from 1-1.5 μm , we can now routinely achieve finesse values at the 300 000 to 700 000 level.

We have discussed these points in the revised manuscript in the Introduction and Methods sections, and provided the citations above.

- If it is the design, i.e., the chosen transmission versus the losses, are the chosen values the best possibility or how would I choose even better?

For cavity ringdown spectroscopy, the optimal choice for coating transmission can only be determined by a model of the entire system, accounting for the required path enhancement (dependent on the optical cross-section of the analyte of interest), laser linewidth (how much of the laser spectrum can be coupled efficiently into the cavity will depend on the cavity finesse), available source power, and the minimum SNR requirements for detection (dependent on the amount of detection noise).

As mentioned in the Introduction, the best-case on-resonance cavity transmission can be calculated as $(T/(T+S+A))^2$, where S and A are scatter and absorption losses, respectively. Therefore, one can calculate the T required for a certain desired minimum power falling on a detector knowing the power and spectral linewidth of a given source.

I believe such a discussion would be interesting and separate the novel aspects of the presented devices/fabrication process over the incremental improvements.

In incorporating the reviewer's comments in the manner described above, we have greatly expanded the scope to encompass novel aspects (such as the hybrid coatings) as well as the incremental improvements (such as process refinement and layer design).

The report is technical and brief, therefore, it requires a level of knowledge of the field higher than I would assume for the general reader of a multidisciplinary journal such as Nature Communications. Nevertheless, when addressing the above point and adding some methodological explanations, I think it can be considered for publication in Nature Communications.

We acknowledge the reviewer's recommendations to present the results in a more general light and apologize for the technical nature of the submitted manuscript. We trust that the modifications and clarifying comments make the paper more accessible to a general audience. Specifically, our introduction is now more conceptually focused, the Design, Fabrication, and Optical Validation section features a discussion of the relative merits (production and performance) of the all-crystalline vs the hybrid designs. Figures 1 and 2 have been replaced to achieve this modification in presentation. Details of the fabrication process and coating design and measurement techniques have been added and moved to the Methods section for completeness.

Minor comments:

1) Fig 1: The top horizontal axis lacks its label.

The FTS plots in Fig. 1 have been replaced by a more general introduction appropriate to a multidisciplinary journal.

2) A visualization of the fabrication process could be helpful. It is currently described but seems to be one of the reasons for the achieved high quality of the mirrors. Therefore, it could deserve a figure.

We have replaced Fig. 1 with a schematic view of the all-crystalline and hybrid layer structure and bonding processes to produce the mirrors. It serves as an introduction to the central concepts of the manuscript and a reference for the layer designs.

3) Fig. 3 shows vertical error bars only for 2 data points. I assume the other error bars are smaller than the data point markers. If this assumption is true, the marker size could be decreased. Also, the authors could comment on why these two data points show much larger uncertainties.

This assumption is true, but decreasing the marker size will not help as the typical uncertainties are of order 0.2 ppm. The caption in Fig. 3 now includes an explanation for the effect: "Wavelengths at extremes of the laser tuning range resulted in lower source power and larger uncertainty in loss."

4) The authors performed cross-sectional SEM to probe layer thicknesses but show no data. I would be interested if any differences are visible between the purely grown and the bonded layers in the middle of the stack.

A detailed description of the as-grown all-crystalline coating is given in Perner 2023, including a discussion of the differences between the 1/8-wave layer to the other layers. We have cited this paper in the section about coating characterization on page 3 and again in the Methods section.

5) I don't understand the reference on page 2: (see Figure 1); v)

The 'v)' refers to the fifth characterization step. The reference to '(see Figure 1)' refers to the text in the fourth characterization step.

6) The paper says the spatially dependent data in Fig. 4 was taken by translating the cavity mirrors with respect to the incoming beam and attributes the observed variations to the bonding processes. Could translation away from the optical axis also cause a misalignment of the cavity? I would appreciate a comment on this aspect.

Since the radius of curvature of the mirrors are known, it is possible to calculate the amount of yaw and pitch necessary to bring the mirror back to normal with respect to the optical axis (defined by the fixed laser). The discussion is clarified by explicitly stating this step: "By successively repositioning the mirrors (and realigning to ensure the curved surface of the mirrors remain normal to the laser), it was possible

to map the optical losses across the coating [13]". Additionally, Ref 13 (Truong 2019) has been included, which details the automated system for performing coating loss measurements over the coating area. Specifically, Ref 13 states that: "For mapping over non-planar substrates of radius-of-curvature R , an additional adjustment of the tip or tilt angle of the mirror is required after translation. Supposing that the mirror normal is initially colinear with the beam axis through the cavity, a translation of distance ρ in the plane orthogonal to the beam will result in the beam now sampling a new part of the mirror with a normal that is deviated by $\theta \approx \rho/R$."

7) The methods section could be subdivided.

Nature Communications encourages the Methods section to be as brief as possible, but this section can be longer as necessary (which we believe is necessary given the reviewer's comments). We are unsure if we should further subdivide and lengthen the Methods section, but will seek editorial advice.

G.D. Cole, *et. al.*, "High-performance near- and mid-infrared crystalline coatings," *Optica* 3, 647-656 (2016)

B. J. Bjork et al., "Direct frequency comb measurement of OD + CO \rightarrow DOCO kinetics," *Science* 354, 444-448 (2016)

G.D. Cole, *et. al.*, "Tenfold reduction of Brownian noise in high-reflectivity optical coatings." *Nature Photon* 7, 644–650 (2013)

L. W. Perner, *et. al.*, "Simultaneous measurement of mid-infrared refractive indices in thin-film heterostructures: Methodology and results for GaAs/AlGaAs," *Phys. Rev. Res.* (2023 in press).

G. W. Truong, *et. al.*, "Near-infrared scanning cavity ringdown for optical loss characterization of supermirrors," *Opt. Express* 27, 19141–19149 (2019).

Reviewer #2 (Remarks to the Author):

The work reported the world-record finesse of 200,000 on a pair of infrared mirrors at 4.45 μm , which is a significant advancement in the mid-infrared band and could improve the performance of cavity ring-down spectroscopy for trace gas sensing by orders of magnitude. It is also impressive to see that the quality and uniformity of the epitaxy-grown coating has been quickly approaching its counterparts in the visible and near-infrared bands. I believe that the significance of this result has reached the standard for publication in Nature Communications, although the technique for mirror fabrication has already been published before. The experimental methods are solid, and the manuscript is well written. Before accepting, I would like to make the following suggestions and comments.

We thank the reviewer for the positive remarks in favor of publication in Nature Communications.

1. In Fig. 2(a), the cavity ring-down traces seem to start from quite different signal levels. Is it because of the fluctuation of the input power to the cavity or something else fluctuating in the setup?

The initial amplitude variation in the optical ringdown signal arises due to fluctuations in the input power, particularly exacerbated by residual amounts of optical feedback to the laser. Fig 2(a) was intended to show that the measured time constant was independent of the initial exponential amplitude (as it should be).

However, as suggested by other reviewers and the editor, we have decided to replace Fig. 2 (as well as Fig. 1) to broaden the scope of interest and focus on the conceptual innovations embodied in the all-crystalline and hybrid coating techniques and how they specifically address longstanding material limitations in the MIR. Instead, Ref 8 (Winkler 2021) has been cited in the discussion of our implementation of a MIR cavity ringdown lossmeter, and Ref 13 (Truong 2019) details the system verification of the particular lossmeter used in this paper.

2. In the first paragraph of page 2, where the coating design has been discussed, can you provide some quantitative information on surface and bulk scatters? For example, surface roughness measurements on top surfaces of the end mirror or the half mirror stack that could provide an estimate on scattering loss.

The typical surface roughness was reported in Ref. 8 (Winkler 2021) and the expected total integrated scatter can be calculated from Ref. 34 to be negligible. In the Methods section, we have added: "Typical RMS surface roughness below $\sigma = 0.2 \text{ nm}$ is achieved [8], which leads to negligible expected scatter levels of order $1 - \text{Exp}(-(4\pi\sigma/\lambda)^2) \approx 0.3 \text{ ppm}$ [34].

3. Relating to the comment above, in the first paragraph of page 3, where the linear absorption coefficient has been calculated, how does the result compare with the absorption coefficient of epitaxy-grown GaAs and AlGaAs films reported in the literature? I think it would be beneficial to compare these results to confirm the absorption is the limiting factor here. And more importantly, is there a path towards even lower excess optical loss of these mirrors?

At one extreme, metal organic chemical vapor deposition (MOCVD) or metalorganic vapor-phase epitaxy (OMVPE) deposited films have typical linear absorption coefficients from $1\text{-}10\text{ cm}^{-1}$. The main limitation in these materials is carbon incorporation from the metal-organic precursors, including trimethylgallium (TMGa), arsine, etc. MOCVD films can be improved slightly via compensation (Pohl 2018), but, even with this optimization, absorption in the NIR remains at the $> 10\text{ ppm}$ level (with MIR films having an absorption a factor of a few higher).

In comparison, MBE grown films, owing to the high purity of the elemental sources and ultra-high vacuum background pressures, enable a significant reduction in the background impurity level. Pushing further, optimized MBE processes have enabled absorption coefficients $< 0.01\text{ cm}^{-1}$ in NIR coatings (for center wavelengths from $1\text{-}1.5\text{ }\mu\text{m}$), corresponding to background doping levels at the $\leq 10^{14}\text{ cm}^{-2}$ (acceptor dominated). This is 1-2 orders of magnitude lower than the best CVD-derived films and represents the state-of-the-art in MBE grown AlGaAs structures.

It is not clear whether this can be improved further as we are likely source-material limited at this level. For additional information outside of the scope of this work I point out the following effort on investigating the limitations in the purity of MBE-grown III-V structures: <https://uh.edu/svec/wsfp-desc.html>

4. As a minor comment, in the third paragraph of Methods, I realize that the refractive indices of the GaAs and AlGaAs layers was measured on a cleaved sample in the same epitaxy run after several reads. Is that correct?

This is correct. We utilize production MBE systems capable of simultaneous deposition on $14\text{ x }4\text{''}$ or $7\text{ x }6\text{''}$ wafers. Thus, we can pull “witness” epi material from the chamber to quantify the material properties including layer thicknesses, absorption / doping (impurity concentration), surface roughness and scatter, etc. In this case we were even more careful, cleaving material from the same epi wafer in order to investigate the as-grown layer structure and quantify the transmission using the following procedure. As mentioned in the Methods section, this process is detailed in Ref. 12 (Perner 2023)

But when first read, the sentence “we modeled the DBR structure using refractive index values measured in-house for the GaAs and AlGaAs” does not seem to specify whether the measurement was done on the bonded mirror or the cleave sample. It would be beneficial to point this out for the general audience.

We have clarified this sentence to read: “...the GaAs and AlGaAs layers using the cleaved sample of epitaxial material grown in the same run as for the completed mirrors. (see [12] for details),...”

G. Winkler, et. al., G. D. Cole, and O. H. Heckl, "Mid-infrared interference coatings with excess optical loss below 10 ppm," *Optica* 8, 686–696 (2021).

G. W. Truong, et. al., "Near-infrared scanning cavity ringdown for optical loss characterization of supermirrors," *Opt. Express* 27, 19141–19149 (2019).

J. Pohl, G. D. Cole, U. Zeimer, M. Aspelmeyer, M. Weyers, "Reduction of absorption losses in MOVPE-grown AlGaAs Bragg mirrors," *Optics Letters*, vol. 43, no. 15, pp. 3522-3525, 1 August 2018.

L. W. Perner, *et. al.*, "Simultaneous measurement of mid-infrared refractive indices in thin-film heterostructures: Methodology and results for GaAs/AlGaAs," *Phys. Rev. Res.* (2023 in press).

Reviewer #3 (Remarks to the Author):

The manuscript "Transmission-dominated mid-infrared supermirrors with finesse exceeding 200 000" by Truong et al. reports the production and careful characterization of high-reflectivity and low-optical-loss mirrors for mid-infrared wavelengths around 4.45 μm .

The paper is well-written and concisely conveys the main results. These results are indeed impressive and bare the potential to advance the applications mentioned in the manuscript, and should therefore be published. However, as it stands, the manuscript would be suitable for publication in a journal specialized in optics rather than in a journal aimed at a broad readership, such as Nature Communications.

The authors could, however, substantially widen the scope of the paper by adding a proof of principle, demonstrating, e.g., that indeed this cavity can improve detection limits for trace gases by orders of magnitude as compared to the state of the art. In this case, I believe that this work could fit to the scope of Nature Communications.

We thank the reviewer for this positive assessment of our paper and acknowledge that it will benefit from the expansion of its scope to appeal the broader readership of Nature Communications. We have expanded the scope with two primary additions to the manuscript: 1) we introduce the "hybrid" coating technique that combines the manufacturability advantages of conventional amorphous coatings with the low-loss MIR quality of the crystalline mirrors to further increase the achieved cavity finesse from just over 200 000 to greater than 400 000, and 2) we expanded our collaboration to include a proof of principle demonstration of the use of the hybrid mirrors in a cavity ringdown spectrometer. In this demonstration, we measured a noise equivalent absorption of $3.8 \times 10^{-9} \text{ cm}^{-1} \text{ Hz}^{-1/2}$ (normalizing to a 10 cm cavity) – representing the lowest noise floor measured in any linear-absorption cavity at wavelengths near 4.5 μm . In an example of trace gas spectroscopy, an ultra-pure 99.999% nitrogen sample was flowed through the cavity and residual amounts of CO and N₂O corresponding to 37 ppb and 141 ppt, respectively, was observed by measuring weak absorption lines in the MIR.

In addition, addressing the following points could strengthen the paper:

1. The claim "... allowing improvements in detection limits by orders of magnitude." from the abstract is not clearly substantiated in this paper. For instance, the improvement in cavity finesse with respect to [5] is "only" a factor of 4. Even though the authors comment that in refs. [5,6] the component losses are not stated, a more conservative claim in the abstract would be more accurate.

We agree that it is difficult to directly demonstrate a spectroscopic system with detection limits orders of magnitude lower than the state-of-the-art. This is because the detection limit of a spectrometer system can be limited by any one of its subsystems e.g., laser linewidth in comparison to the cavity linewidth, available source power, mirror losses, sample preparation and gas flow handling, and detector noise. However, the quality of MIR mirrors and coating losses are currently the limiting factor in state-of-the-art experiments, and our paper focuses on the advancements we can offer with true MIR supermirrors, in this case our all-crystalline and hybrid coatings. It would take substantial efforts and funding to acquire the state-of-the-art performance in all other sub-systems in order to for us to realize

the best spectrometer, but this is an undertaking that is distinct from this manuscript's report of the best MIR mirrors manufactured to date.

Nevertheless, we agree that addressing this point strengthens the paper. We have therefore substantially revised the introduction, removing the claim of the expected scaling of detection limit based solely on the achieved finesse and replacing it with a statement of our achieved noise performance in the proof-of-concept demonstration: "...we achieve the lowest noise-equivalent absorption in a linear cavity ring-down spectrometer normalized by cavity length."

2. The bandwidth of the mirrors could be described more precisely in the text of the paper.

We are unaware of a standard metric for describing the bandwidth of ultra-low loss mirrors since it depends on the maximal losses tolerable for a given system. We therefore provide plots of the mirror losses as a function of wavelength (Fig. 3). In addition, we note that the hybrid mirrors do have a slightly wider stopband than the all-crystalline mirrors due to the higher refractive index contrast of the amorphous substack layers (page 3).

3. On page 2, I do not understand the sentence "Mirror die were defined via lithography and etching..."

In microfabrication, a die refers to the production of smaller pieces from a parent wafer. In this case, the final coatings (~ 8 mm diameter) were chemically cut from a 15 cm wafer using lithography and etching. Further information on the microfabrication is now provided in the Methods section, additional detail can be found in Refs 10 (Cole 2016) and 8 (Winkler 2021).

4. Importantly, for a paper addressing a broad audience, the state of the art should be presented in a broader context. The authors cite only 18 references, 7 of which are to their previous work. It would improve the quality of the paper if they could, e.g., give an overview of resonators demonstrated in the mid-infrared spectral region.

In addition to the references mentioned, the revised manuscript includes an overview of resonators in the context of cavity ring-down experiments reported in the literature operating at wavelengths near 4.5 μm (refs 7, 15-17, 19-23, 25, 26). In particular, Table 1 grounds our achieved detection limit in the proof-of-concept demonstration of a linear-absorption cavity ring-down in the context of equivalent state-of-the-art instruments and find that we have the lowest noise-equivalent when normalized by cavity length, which is a direct result of the coating improvements.

Reviewer #4 (Remarks to the Author):

The paper by Truong and co-workers introduces for the first time mirrors with extremely high finesse at mid-infrared wavelengths, overcoming the well-known reflectivity and loss limitations in this wavelength range. The result obtained, namely a finesse of 230000 at 4.45 μm with only 4 ppm losses, corresponding to a 45% cavity throughput under optimal conditions, sets for these mirrors a quality on a par with near-infrared mirrors. In the long term, this is likely to produce important advances in precision spectroscopy and ultra-sensitive gas detection.

It is very difficult to decide whether this paper deserves or not a publication in Nature Communications: on the one hand the quality of the result and its scientific relevance, particularly if these mirrors could be made commercially available in a future for applications, would definitely justify a high-impact factor publication. On the other hand, this paper does not contain any novelty as compared to the Optica paper published two years ago by the same authors: the technology is the same, the methods are nearly the same, the mirror excess losses are just less than a factor of 2 better (4 ppm rather than 7 ppm). What changes is the number of periods in the mirror stack, increased from 34 to 44. This technical step, though it was not obvious, was already well within the expectations of anyone that had read the first paper (even the magic number of periods, 44, was already anticipated in the conclusions of the Optica paper). I let the Editor take a final decision on the basis of this and other reviews.

We thank the reviewers for the positive comments recognizing the profound implications of realizing a $>200\,000$ finesse cavity in the MIR. However, the reviewer requests clarity in the conceptual advancement of this achievement beyond Ref 8 (Winkler 2021), and path towards commercial viability.

To address these concerns, and similar concerns from other reviewers, we have taken the editor's invitation for major revisions. We now introduce the "hybrid" coating technique that combines the manufacturability advantages of conventional amorphous coatings with the low-loss MIR quality of the crystalline mirrors to further increase the achieved cavity finesse from above 200 000 to above 400 000. This new paradigm of optical coating that combines amorphous and crystalline deposition processes, the first to be reported in the scientific literature, clearly demonstrates conceptual advancement that simultaneously provided commercial viability and even higher cavity finesse! We discuss these points, comparing the scalability of production processes on page of the revised manuscript, and show the lowered losses and uniformity in Figs. 3 and 4, respectively.

We are confident that these additions and improvements beyond what the reviewer previously saw will only solidify their conviction that such results are deserving of publication in Nature Communications, subject to the minor clarifications implemented as a result of the comments below.

If this paper is accepted for publication, I just encourage the authors to better clarify these points:

- *The total losses measured with cavity-ring down spectroscopy are given with error bars that seem to be overestimated. For example, at the optimal wavelength of 4.45 μm , the total losses measured by averaging 500 ring-down signals are said to be 13.6 ppm \pm 0.49 ppm, which is equivalent to an uncertainty of 3.6%. This would imply, for a single ring-down event, an uncertainty as large as 3.6% * (500)^{0.5} = 80%. The authors should then better comment the origin of their error bars.*

We have been conservative and chosen to state the standard deviation of the ensemble at each wavelength. The reviewer has assumed the stated error is the standard error of the mean. To avoid confusion, the caption to Fig 3 new includes: "Vertical error bars represent the standard deviation of the ensemble of measurements taken at a particular wavelength... Wavelengths at extremes of the laser tuning range resulted in lower source power and larger uncertainty in loss."

- The cavity-ring-down traces in Fig.1 show a pretty poor signal-to noise ratio because the peak-to-peak residuals (10 mV, panel a) are only 10-15 times below the maximum signal (100-150 mV, again panel a). Why is the signal-to-noise ratio so poor ?

We assume the reviewer refers to Fig. 2 in the original manuscript (Fig. 1 was a broadband FTS spectrum of the mirror stopband). The SNR was limited by the amount of light we could collect on the detector given the narrow cavity linewidth (4.5 kHz for the all-crystalline and 2.5 kHz for the hybrid mirrors at 4.45 μm when tested in the 145 mm cavity length configuration). As described in the manuscript, we employed a free-running extended-cavity FP-QCL (with a typical emission bandwidth of $\gg 1$ nm) which leads to a significant spectral mismatch between the laser and the cavity under test. In addition, the overall measurement is subject to losses due to polarization mismatch and monochromator grating efficiencies, leading to the final observed SNR. However, critically, we note that this SNR was sufficient to resolve $\ll 1$ ppm of changes in loss as evidenced by the typical sample standard deviation at each wavelength point of Fig. 3.

Note that we have needed to replace Figs. 1 and 2 in the resubmitted manuscript due to the editorial request for major revisions to demonstrate conceptual advancement beyond previous MIR crystalline coating work. Since the achieved SNR was inconsequential to the measurement of losses as a function of wavelength, we do not believe this omission impacts the quality of the results presented.

For the completeness of this review, we show a comparison of

- i) Fig. 2 from the original manuscript, which refers to the losses of the all-crystalline mirrors at 4.45 μm taken on the 145 mm coating lossmeter,
- ii) Ring-downs taken for the hybrid mirrors also taken on the 145 mm coating lossmeter, and
- iii) A typical ring-down taken for the hybrid mirrors installed in the empty 79 cm CO/N₂O spectrometer.

The 79-cm spectrometer used a higher spectral brightness sources and lower noise (cooled InSb) detectors and amplifiers. With > 10 -fold increase in the single-ring-down SNR, we were able to confirm the same coating losses. A small variation is attributable to residual coating non-uniformity (Fig 4 in manuscript) and significant change in spot size and location when the mirrors were changed between experiments.

Top Left: Fig. 2 taken from the original manuscript showing consecutive ring-downs taken at a particular location on the all-crystalline mirrors using the 145-mm coating lossmeter. SNR ~ 40 (sdev of residuals: 2.5 mV). Top Right: consecutive ring-downs taken at a particular location on the hybrid mirrors using the 145-mm coating lossmeter. SNR ~ 20 (sdev of residuals: 2.3 mV). Bottom: typical ring-down taken at a particular location on the hybrid mirrors using the empty 79-cm spectrometer. SNR ~ 175 . The small difference in the measured value of $1-R$ of the hybrid mirrors taken from the 145-mm coating lossmeter and in the 79-cm spectrometer is due to difference in experimental conditions (such as spot size and location).

- The number of ring-down signals N that has been used to determine the mirror losses is strangely kept very high around $4.45 \mu\text{m}$ where the ring-down time and thus the SNR is maximum (N about 500), while being kept low at the edges of the mirror reflection band ($N=60$ at $4.33 \mu\text{m}$, when the ring-down time is

10 times smaller). Shouldn't one choose the opposite approach, with a higher N at the edge of the reflection bandwidth?

It was the original intent to keep N constant across all points in Fig. 2. However, we state in the text: "Fewer valid ring-downs were obtainable towards the extremes of the laser tuning range, and the uncertainties increased due to lower signal-to-noise." The tuning range of the laser was ultimately limited by the ability to control the optical feedback from the external cavity relative to the bare laser gain dynamics.

- I would like to have a comment on the spectral width of the reflection band: this is actually pretty narrow if one considers that the reflection drops by a factor of 10 at 4.33 and 4.58 μm , which are only 5% away from the maximum reflectivity point. How could one enlarge the reflection band ?

We acknowledge the narrow bandwidth of the all-crystalline coatings, and this is unavoidable due to the limited index contrast between the GaAs and AlGaAs materials. The high refractive index GaAs layers exhibit n_{GaAs} of 3.28, while for the low index AlGaAs, $n_{\text{AlGaAs}} = 2.87$, which compares unfavorably to Si/SiO₂ stacks with 3.50/1.45 (see Fig. 2 in the revised manuscript). This limited index contrast leads to inherently narrowband reflectors for quarter-wave multilayers. This could be expanded by designing non-quarter-wave interference coatings, for example chirped or multiband mirrors. We acknowledge that we are trading off (ultralow) optical losses for bandwidth in these structures. In the Methods section, we note that: "Additionally, advances in epitaxy exploiting alternative material platforms could enable wider bandwidth mirrors via enhanced index contrast [Schön 2000], however, it remains to be seen whether such materials system will be capable of sufficiently low optical loss." In the discussion of the mirror tradeoffs, we note that Fig. 3 shows "a wider stopband (due to the higher refractive index contrast of the amorphous layers)" for the hybrid mirrors.

S. Schön, M. Haiml, U. Keller, "Ultrabroadband AlGaAs/CaF₂ semiconductor saturable absorber mirrors", Appl. Phys. Lett. 7 August 2000; 77 (6): 782–784. <https://doi.org/10.1063/1.1306917>

REVIEWER COMMENTS

Reviewer #1 (Remarks to the Author):

Second Review for "Transmission-dominated mid-infrared supermirrors with finesse exceeding 200 000" by Truong et al. (2022), now "Mid-infrared supermirrors with finesse exceeding 400 000"

In their new manuscript, Truong et al. report on their initial supermirror design and a second, hybrid, supermirror design to achieve high-quality mirrors at 4.45 μm central wavelength. They also added a proof-of-principle experiment showing absorption spectroscopy in high-purity nitrogen. I commend the authors for the large effort they made to improve the general appeal of the manuscript. The hybrid design adds novelty and, judging from the data, is an improvement over the old manufacturing process, adding an argument for publication of the manuscript in Nature Communications, which I recommend.

Points that must be remedied:

1. Fig. 5: The x-axis in panel c makes no sense, some error must have occurred in figure preparation. The dashed line is labeled $t^{-1/2}$, which seems to lack a constant. Maybe it is $\alpha_0 * t^{-1/2}$?

Some points that would help the presentation:

1. A broadband transmittance or reflectance calculation for both mirror designs could reappear (it existed in Fig. 1 of the initial manuscript), possibly as supplementary material.
2. If the material/mirror examined in Perner et al. 2023 is the exact material/mirror examined in this publication (as seems from the rebuttal), the manuscript should state this.
3. The cavity ringdown raw data (former Fig. 2) could re-appear, possibly as supplementary material.
4. Fig. 2: No unit is given for $|E|^2$, which is okay, but it could state arbitrary units. Better, even, the right y-axis could be employed to plot a normalized $|E|^2$. The colors used for a-Si and Al₂O₃ are indiscernible. The lower panel lacks axes labels.
5. Fig. 4: The color map is chosen in a way that reveals no detail in panel b. A second color map could be used instead to add information.
6. Pg. 3: The reason for the ability to extend the center wavelength beyond 12 μm could be explained with a sentence.
7. The discussion of the Allan deviation could be expanded, it is very brief, and the text does not mention the fit nor slope plotted in Fig. 5. If α_0 is a slope, then the caption of Table 1 shouldn't call it Allan deviation. The slope would fit the unit in the table ($\text{cm}^{-1} \text{Hz}^{-1/2}$). However, it seems to me it would not have to be measured at 1 second specifically if it is a slope. This should be harmonized.
8. Pg. 4: Second to last sentence, "compact and atmospheric trace gas" seems to lack a word.

Reviewer #2 (Remarks to the Author):

In this revised manuscript of "Mid-infrared supermirrors with finesse exceeding 400 000", the authors presented a hybrid "amorphous + crystalline" fabrication flow of infrared mirrors and set a new finesse record twice as high as in the originally reported "all-crystalline" approach. The authors then built an optical cavity with these mirrors and demonstrated trace gas detection in ultra-high purity nitrogen.

The idea of bonding the crystalline coating to the amorphous one is quite neat and novel. The crystalline GaAs/AlGaAs multilayer sets the low-loss property of the full coating in the mid-infrared band, while the amorphous a-Si/SiO₂ with its high index contrast provides flexibility and versatility in transmissivity and bandwidth design. And the authors experimentally showed that the combination of the two indeed combines their benefits. This whole approach could also benefit the manufacturability of the crystalline mirrors, as it now only involves one bonding process, and to me, it makes full use of the low-loss property of the crystalline mirror.

The demonstration of trace gas detection showcased an application of these low-loss mirrors, and the fact that the cavity can detect trace gases at ppb level is impressive. When compared with other works, in the absolute sense, this system is on par with the state-of-the-art one in terms of the noise-equivalent absorption. After factoring in the length of the cavity, this system currently outperforms all the other linear cavity ring-down spectroscopy systems.

In conclusion, the authors expanded on the original work and further advanced the technology of supermirrors in the mid-infrared band. In the rebuttal letter, authors also addressed all my previous questions and concerns. Same as in my previous recommendation, I believe that the significance of this result has reached the standard for publication in Nature Communications, and now with more novelty. I would recommend accepting this manuscript, and the comments below are minor.

1. Why do you use a 1/8 Al₂O₃ layer to cap the amorphous layers? Is it related to the surface adhesion for bonding purposes?
2. It seems the ticks on the time axis of fig. 5(c) are mislabeled.
3. In the last part of section "Applications in gas sensing and spectroscopy" where the authors compare the performance of their system with others, they claim that "the noise equivalent absorption, α at 1 s, is inversely proportional to the square of the cavity length." The authors should consider briefly explaining this with equations, perhaps in methods or supplementary information, as this is the main reason that the performances of different systems should be normalized with respect to the cavity length. Curious readers outside this field would be interested to learn more on the performance of the cavity ring-down spectroscopy without tracing back too many references.

Reviewer #3 (Remarks to the Author):

The authors have carefully considered all points raised by myself and the other referees. In particular, they have added a proof-of-principle measurement, directly and unequivocally showing the performance of their novel mirrors in cavity-ringdown spectroscopy. In the present form, I recommend publication of the manuscript in Nature Communications.

Reviewer #4 (Remarks to the Author):

The authors have thoroughly revised their manuscript and they have substantially addressed all weak points and requests raised by reviewers. In its current form it is in my view susceptible for publication in Nature Communications.

There is a mistake in the horizontal axis of Fig. 5c: the time scale is increasing from 10^{-2} to 0 and all of a sudden decreasing back to 10^{-2} . Moreover, the minimum of Allan deviation is not at 8s as reported in the text. This should be fixed.

Reviewer #1 (Remarks to the Author):

Second Review for “Transmission-dominated mid-infrared supermirrors with finesse exceeding 200 000” by Truong et al. (2022), now “Mid-infrared supermirrors with finesse exceeding 400 000”

In their new manuscript, Truong et al. report on their initial supermirror design and a second, hybrid, supermirror design to achieve high-quality mirrors at 4.45 μm central wavelength. They also added a proof-of-principle experiment showing absorption spectroscopy in high-purity nitrogen. I commend the authors for the large effort they made to improve the general appeal of the manuscript. The hybrid design adds novelty and, judging from the data, is an improvement over the old manufacturing process, adding an argument for publication of the manuscript in Nature Communications, which I recommend.

We thank the reviewer for their positive feedback on our re-submitted manuscript.

Points that must be remedied:

*1. Fig. 5: The x-axis in panel c makes no sense, some error must have occurred in figure preparation. The dashed line is labeled $t^{-1/2}$, which seems to lack a constant. Maybe it is $\alpha_0 * t^{-1/2}$?*

There was an error in generating the plot in Fig 5c. This has been corrected and the function for the $t^{-1/2}$ trendline has been explicitly provided in the plot.

Some points that would help the presentation:

1. A broadband transmittance or reflectance calculation for both mirror designs could reappear (it existed in Fig. 1 of the initial manuscript), possibly as supplementary material.

This and the cavity ringdown raw data have re-appeared as supplementary material for completeness. We also included a Reference to this supplementary material in the Methods section of the revised manuscript.

2. If the material/mirror examined in Perner et al. 2023 is the exact material/mirror examined in this publication (as seems from the rebuttal), the manuscript should state this.

The materials examined in Perner et. al. 2023 comprised a “cleaved sample of epitaxial material grown in the same run as for the completed [all-crystalline] mirrors.” However, while taken from the exact same growth run, the wafer material analyzed in Perner 2023 was from a geometrically different part of the epilayer material. As explained in our Methods section, this will lead to a shift in center wavelength while other material properties are virtually identical. “For the hybrid mirrors, the crystalline stack came from the same epi wafer as employed for the all-crystalline mirrors...” These statements were made in the Methods section of the submitted manuscript.

3. The cavity ringdown raw data (former Fig. 2) could re-appear, possibly as supplementary material.

This and the broadband transmittance have re-appeared as supplementary material for completeness. We also included a Reference to this supplementary material in the Methods section of the revised manuscript.

4. Fig. 2: No unit is given for $|E|^2$, which is okay, but it could state arbitrary units. Better, even, the right y-axis could be employed to plot a normalized $|E|^2$. The colors used for a-Si and Al₂O₃ are indiscernible. The lower panel lacks axes labels.

The right y-axis of Figure 2 now shows normalized $|E|^2$ in arbitrary units. Axes labels have been repeated on the lower panel for clarity. The color used for Al₂O₃ has been changed to make it discernably different from a-Si.

5. Fig. 4: The color map is chosen in a way that reveals no detail in panel b. A second color map could be used instead to add information.

A second color map has been added for panel b, revealing (with greater loss resolution) that the hybrid mirror coating shows excellent uniformity of losses across the central coating area.

6. Pg. 3: The reason for the ability to extend the center wavelength beyond 12 μm could be explained with a sentence.

The wavelength coverage has a fundamental limit, which is the transparency window of GaAs and AlGaAs, which reaches to the reststrahlen region (20 μm , as already stated in the current manuscript). However, as stated in the Methods section, due to limitations in MBE, we are limited to a “maximum epitaxial growth thickness of $\sim 16 \mu\text{m}$ for maintaining a suitably low defect density.” Together with the design wavelength, this will limit the total number of crystalline quarter wave layers. While such HR coatings are certainly feasible, achievable performance will depend on many factors, such as available amorphous materials and deposition methods.

7. The discussion of the Allan deviation could be expanded, it is very brief, and the text does not mention the fit nor slope plotted in Fig. 5. If α_0 is a slope, then the caption of Table 1 shouldn't call it Allan deviation. The slope would fit the unit in the table ($\text{cm}^{-1} \text{Hz}^{-1/2}$). However, it seems to me it would not have to be measured at 1 second specifically if it is a slope. This should be harmonized.

The sloping dashed line on Fig 5c is a trendline with a $t^{-1/2}$ functional dependence. To clarify this point, it is now explicitly labeled in the plot as “ $(6.0 \times 10^{-11}) \times (t^{-1/2})$ ”, and the following explanation of the significance has been added to the text on page 4: “For timescales below 8 s, the Allan deviation approximately follows a $t^{-1/2}$ dependency (dashed trendline), indicating that measurements were dominated by white noise, before being affected by drifts.”

In addition, we have changed the sentence “Separately, we measured the Allan deviation of the empty-cavity time constants (τ_0)...” to “Separately, we measured the Allan deviation of the empty-cavity absorption coefficients (α_0)...” to correctly describe the plot in Fig 5c.

8. Pg. 4: Second to last sentence, “compact and atmospheric trace gas” seems to lack a word.

This sentence now reads: “...compact and atmospheric trace gas detection devices...”.

Reviewer #2 (Remarks to the Author):

In this revised manuscript of “Mid-infrared supermirrors with finesse exceeding 400 000”, the authors presented a hybrid “amorphous + crystalline” fabrication flow of infrared mirrors and set a new finesse record twice as high as in the originally reported “all-crystalline” approach. The authors then built an optical cavity with these mirrors and demonstrated trace gas detection in ultra-high purity nitrogen.

The idea of bonding the crystalline coating to the amorphous one is quite neat and novel. The crystalline GaAs/AlGaAs multilayer sets the low-loss property of the full coating in the mid-infrared band, while the amorphous a-Si/SiO₂ with its high index contrast provides flexibility and versatility in transmissivity and bandwidth design. And the authors experimentally showed that the combination of the two indeed combines their benefits. This whole approach could also benefit the manufacturability of the crystalline mirrors, as it now only involves one bonding process, and to me, it makes full use of the low-loss property of the crystalline mirror.

The demonstration of trace gas detection showcased an application of these low-loss mirrors, and the fact that the cavity can detect trace gases at ppb level is impressive. When compared with other works, in the absolute sense, this system is on par with the state-of-the-art one in terms of the noise-equivalent absorption. After factoring in the length of the cavity, this system currently outperforms all the other linear cavity ring-down spectroscopy systems.

In conclusion, the authors expanded on the original work and further advanced the technology of supermirrors in the mid-infrared band. In the rebuttal letter, authors also addressed all my previous questions and concerns. Same as in my previous recommendation, I believe that the significance of this result has reached the standard for publication in Nature Communications, and now with more novelty. I would recommend accepting this manuscript, and the comments below are minor.

We thank the reviewer again for the positive remarks and for reiterating their recommendation for publication in Nature Communications.

1. *Why do you use a 1/8 Al₂O₃ layer to cap the amorphous layers? Is it related to the surface adhesion for bonding purposes?*

The 1/8-wave Al₂O₃ layer was indeed originally intended to be an adhesion layer to aid bonding. However, we also deposited a version using a 1/8-wave layer of a-Si which worked just as well. On page 2, we now state: “This termination layer was intended to aid bonding, but subsequent designs ending the amorphous stack with a-Si were found to behave identically.”

2. *It seems the ticks on the time axis of fig. 5(c) are mislabeled.*

There was an error in generating the plot in Fig 5c. This has been corrected.

3. *In the last part of section “Applications in gas sensing and spectroscopy” where the authors compare the performance of their system with others, they claim that “the noise equivalent absorption, α at 1 s, is*

inversely proportional to the square of the cavity length.” The authors should consider briefly explaining this with equations, perhaps in methods or supplementary information, as this is the main reason that the performances of different systems should be normalized with respect to the cavity length. Curious readers outside this field would be interested to learn more on the performance of the cavity ring-down spectroscopy without tracing back too many references.

At the location of that statement, we have added: “...according to $\alpha_0 = (1/(c\tau_0^2))\sigma_{\tau_0}$, where c is the speed of light, $\tau_0 = \mathcal{F}L/(c\pi)$ is the empty cavity decay time, \mathcal{F} is the empty cavity finesse and σ_{τ_0} is the statistical error on τ_0 ”.

Reviewer #3 (Remarks to the Author):

The authors have carefully considered all points raised by myself and the other referees. In particular, they have added a proof-of-principle measurement, directly and unequivocally showing the performance of their novel mirrors in cavity-ringdown spectroscopy. In the present form, I recommend publication of the manuscript in Nature Communications.

We thank the reviewer for this positive re-assessment of our paper. We are grateful for the suggestion to expand its scope to appeal the broader readership of Nature Communications, which has resulted in a much-improved manuscript.

Reviewer #4 (Remarks to the Author):

The authors have thoroughly revised their manuscript and they have substantially addressed all weak points and requests raised by reviewers. In its current form it is in my view susceptible for publication in Nature Communications.

We thank the reviewers for the positive comments and for the suggestions that have greatly improved this manuscript.

There is a mistake in the horizontal axis of Fig. 5c: the time scale is increasing from 10^{-2} to 0 and all of a sudden decreasing back to 10^{-2} . Moreover, the minimum of Allan deviation is not at 8s as reported in the text. This should be fixed.

There was an error in generating the plot in Fig 5c. This has been corrected, but none of the reported values in the text were affected by the error.

REVIEWERS' COMMENTS

Reviewer #1 (Remarks to the Author):

Third review for Truong et al., Mid-infrared supermirrors with finesse exceeding 400 000, 2023

The authors have successfully addressed all points and I recommend publishing the article.

Reviewer #2 (Remarks to the Author):

The authors have addressed all the points raised by myself and other referees. In the current form, I recommend the publication of the manuscript in Nature Communications.